# Immunoinformatic analysis for identifying immunogenic antigens from the complete proteome of *Giardia lamblia*

**David Ortega-Tirado[1], Carlos A Velazquez-Valdez[2], Thania Garzon[1], Leslie Bracamontes-Picos[1], Gloria Lopez-Romero[1], Carlos Velazquez[1]/+**

[1]Universidad de Sonora, Department of Chemistry-Biology, Hermosillo, Sonora, Mexico
[2]Universidad de Sonora, Department of Mathematics, Hermosillo, Sonora, Mexico

**BACKGROUND** *Giardia lamblia* is a parasite that infects humans. To date, there is no vaccine available for human giardiasis. Thus, discovering new immunogenic antigens is crucial for the rational design of a vaccine.

**OBJECTIVES** This study aimed to identify the main immunogenic antigens of *G. lamblia* from its entire proteome using immunoinformatic and data science techniques. To our knowledge, this is the first study to systematically identify immunogenic antigens of *G. lamblia* across its complete proteome, providing a comprehensive map of potential immunogenic antigens.

**METHODS** Briefly, FASTA sequences of *G. lamblia* isolates WB and GS were submitted to the NetMHCII 2.3 predictor. The analysis was conducted for five murine major histocompatibility complex (MHC)-II molecules: I-A$^b$, I-A$^d$, I-E$^d$, I-A$^k$, and I-E$^k$. Python 3.9 was used to develop custom code for data processing and analysis.

**FINDINGS** We identified 414 potential immunogenic polypeptides for isolate WB and 350 for isolate GS. For both isolates, most polypeptides contained peptides with high affinity for I-A$^b$ and I-E$^k$. Notably, no polypeptides with high affinity for I-A$^k$ were detected. Homologous potential immunogenic antigens (129 polypeptides) were identified in both isolates. The analysis revealed that 12 potential immunogenic polypeptides from isolate WB and 10 from isolate GS are part of the *Giardia* secretome. Additionally, promiscuous polypeptides that bind to at least two different MHC-II molecules were found in both isolates.

**MAIN CONCLUSIONS** These findings lay a valuable foundation for the rational development of a vaccine against human giardiasis and show a computational strategy that can be applied to the study of other pathogens.

Key words: T cell epitopes - *Giardia lamblia* - immunogenic polypeptide - MHC-II

*Giardia lamblia* is a protozoan parasite that infects the upper intestinal tract of humans and other mammals.[1] This parasite has a life cycle consisting of an infective form known as a cyst and a vegetative form called a trophozoite.[2,3] Infection begins when a host ingests water or food contaminated with cysts. Once in the stomach, the excystation process begins, and each cyst generates two trophozoites. The trophozoites colonise the small intestine without invading the epithelia. Eventually, they migrate to the lower intestinal tract, where environmental changes induce the encystation process, and the cysts are then released through the faeces, completing the life cycle.[2,3]

*Giardia lamblia* is divided into eight assemblages, A-H.[4,5] Assemblages A and B are the main ones responsible for human giardiasis.[4,5] To date, there is no vaccine against human giardiasis. The discovery of new immunogenic antigens of *Giardia* therefore remains pivotal for the rational design of a vaccine against this disease.

Several *Giardia* antigens have confirmed immunogenicity. Some of these are structural molecules, such as α-giardins and α- and β-tubulins.[6] Metabolic enzymes, including fructose-1,6-bisphosphate aldolase (FBA), ornithine carbamoyltransferase (OCT), arginine deiminase (ADI), and enolase, are also sources of immunogenic and conserved molecules.[7]

Surface antigens, predominantly variant-specific surface proteins (VSPs), constitute a group of *Giardia* proteins with high immunogenic properties.[8] Even in the cyst stage, immunogenic molecules such as cyst wall proteins CWP1, CWP2, and CWP3 are present. In addition, the excretory-secretory products of *Giardia*, which shape the parasite's secretome, contain immunogenic proteins, including cathepsins, VSPs, metabolic enzymes such as ADI, OCT, and enolase, and structural proteins such as α-1 giardin, α-2 giardin, and α-11 giardin.[6,9] These *Giardia* antigens can trigger the immune response necessary to control infection. It is well established that the adaptive immune response is essential not only for controlling but also for clearing *Giardia* infection. CD4$^+$ T cells are pivotal for parasite elimination, as previous studies have shown that CD4$^+$-deficient mice are unable to clear *Giardia* infection.[10]

Financial support: This study was supported by Secretaria de Ciencia, Humanidades, Tecnologia e Innovacion (awards numbers CB2017-2018 A1-S-21831 and CBF-2025-I-4042).

+ Corresponding author: velaz2@unison.mx | ⓘ https://orcid.org/0000-0002-3728-7848

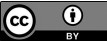

These cells are also relevant in human parasitic infections, as observed in patients with HIV infection. Studies have revealed a higher prevalence of parasites such as *Cryptosporidium* spp. and *G. lamblia* among human immunodeficiency virus (HIV)-infected children with low CD4[+] T-cell counts.[11] Similarly, in adult patients with HIV infection, *G. lamblia* is the most frequently identified parasite, followed by *Entamoeba histolytica* and *Ascaris lumbricoides*.[12,13] Accordingly, the search for new immunogenic antigens could be directed towards those capable of activating CD4[+] T cells.

Despite progress in the identification of immunogenic antigens of this parasite, our understanding of the full repertoire of potential immunogenic proteins encoded in the genome of *G. lamblia* remains limited.

The classical route for antigen discovery begins with the cultivation of the target pathogen under laboratory conditions, followed by its dissection into individual components, the identification and isolation of each component, and the evaluation of the immunogenic capacity of the antigens.[14,15] Subsequently, selected antigens are produced on a large scale to initiate vaccine development. This classical workflow is costly and time-consuming and may also pose biological hazards due to pathogen manipulation and culture.

Nowadays, immunoinformatics represents a faster and more accessible approach for uncovering immunogenic proteins. This technique starts from the pathogen's genome and combines various computational tools to identify immunogenic proteins based on the recognition of peptides by B and/or T cells.[15] For the identification of peptides recognised by CD4[+] T cells, different prediction algorithms are used to identify major histocompatibility complex (MHC)-binding sequences within protein antigens. This approach serves as an alternative strategy for the discovery of new immunogenic antigens in *Giardia* and other pathogens.

Based on the above considerations, the present study aimed to apply immunoinformatics approaches to identify all potential immunogenic proteins from the complete proteome of *G. lamblia*. Selection was based on the identification of T-cell peptides with affinity for MHC class II molecules using the NetMHCII 2.3 tool from the immune epitope database (IEDB). Python 3.9 was also used to process and analyse all derived data. The source of the *G. lamblia* proteomes was GiardiaDB, a repository containing genome and proteome data from several *Giardia* genotypes, encompassing all proteins associated with this parasite. Some data have been validated using microarrays and mass spectrometry, and deprecated genes have been removed.[16] GiardiaDB also contains computationally predicted protein sequences, which may include redundancies or unvalidated entries, thereby highlighting the potential for further investigation.

## MATERIALS AND METHODS

*Data acquisition* - FASTA files containing the full proteomes of *G. lamblia* isolates WB and GS were downloaded from the GiardiaDB repository[17] (https://giardiadb.org/giardiadb/app). In addition, XLSX files containing general information on both proteomes were obtained, including amino acid length, molecular weight, the presence of transmembrane domains, and whether the polypeptides are associated with the trophozoite or cyst stage. It is worth noting that GiardiaDB includes computationally predicted protein sequences, which provides opportunities for further validation in downstream analyses.

*MHC-II peptide binding prediction* - The analysis was conducted between February 2023 and January 2024. Predictions were performed using the NetMHCII 2.3 tool, accessible via the IEDB (www.iedb.org) RESTful API.[18,19] Implementation was carried out using the Python SDK for the IEDB API Tools to facilitate interaction with the API for peptide-binding prediction. Predictions were performed against five murine MHC class II molecules: I-A$^k$, I-E$^k$, I-A$^d$, I-E$^d$ and I-A$^b$. The output from the MHC class II peptide-binding predictions was merged with general polypeptide information. This integration enabled a comprehensive analysis, combining binding affinity predictions with general features of each polypeptide. A polypeptide was considered immunogenic if at least one of its T-cell peptides bound to any of the MHC class II molecules described above.

Once the data were merged, a general analysis was conducted for the different percentile ranks available. This step provided an overview of peptide-binding affinities across the proteome, allowing the identification of polypeptides with high binding potential. For a more precise analysis, the data were subsequently filtered to include only polypeptides with T-cell peptides exhibiting a percentile rank $\leq 0.01$, an IC$_{50}$ value $\leq 50$ nM, and a protein length $\leq 3,000$ amino acids. These filtering criteria ensured the selection of polypeptides with the highest binding affinity within biologically relevant polypeptide length limits.

*Identification of immunogenic polypeptides in the secretome of G. lamblia assemblages A and B* - To identify all immunogenic polypeptides in the secretomes of the GS and WB isolates of *G. lamblia*, the filtered MHC class II prediction data were merged with the XLSX files obtained from the study by Ma'ayeh and colleagues.[9] The XLSX files used for the merge were titled: "S1 Table. All proteins identified in the secretome of Giardia intestinalis WB isolate in serum-free RPMI-1640 medium" and "S2 Table. All proteins identified in the secretome of Giardia intestinalis GS isolate in serum-free RPMI-1640 medium".

*Homology analysis of the immunogenic polypeptides of G. lamblia* - The filtered MHC class II prediction data were subjected to homology analysis. Specifically, polypeptides from *Giardia* Assemblage B isolate GS and *Giardia* Assemblage A isolate WB were analysed to identify those conserved in both isolates. Sequence similarity was calculated using the Biopython module, which performs pairwise sequence alignment via a dynamic programming algorithm. Polypeptides with a sequence identity $\geq 80\%$ were considered homologous.

*Analysis of the interactions of the immunogenic T-cell peptides of G. lamblia with MHC-II molecules* -

We conducted a literature search to identify the structures of the MHC class II molecules analysed in this study: I-A$^d$, I-E$^k$, and I-A$^k$.[20,21,22]

The anchor residues of model peptides bound to I-A$^d$, I-E$^k$, and I-A$^k$ were compared with the most immunogenic T-cell peptides from the WB and GS isolates. Based on the filtered MHC class II binding prediction data, T-cell peptides with the lowest IC$_{50}$ values were selected as the most immunogenic for both *G. lamblia* isolates.

## RESULTS

*Immunogenic polypeptides identification by using immunoinformatic and the whole G. lamblia proteome* - The rational design of vaccines involves the identification of immunogenic antigens of the pathogen. In this study, our objective was to identify immunogenic polypeptides from the entire proteome of *G. lamblia*. We used the proteomes of the WB isolate (assemblage A) and GS isolate (assemblage B) from GiardiaDB to predict T-cell peptides with the highest affinity for MHC class II molecules. Out of 4,970 polypeptide chains (rep-

resenting 100% of the *G. lamblia* WB proteome) submitted for prediction, a total of 414 chains (8%) were identified as immunogenic (percentile rank ≤ 0.01 and IC$_{50}$ ≤ 50 nM) for the WB isolate (Figure A). Of these, 33% were associated with I-E$^k$, 31% with I-A$^b$, 23% with I-A$^d$, and 13% with I-E$^d$. No immunogenic polypeptides were identified for I-A$^k$ under the selected parameters.

Out of a total of 4,470 polypeptide sequences (representing 100% of the GS isolate proteome), 350 chains (8%) were identified as immunogenic (Figure B). Among these, 36% of the polypeptides were bound to I-E$^k$, 31% to I-A$^b$, 17% to I-A$^d$, and 16% to I-E$^d$. No immunogenic polypeptides with affinity for I-A$^k$ were identified under the selected parameters (percentile rank ≤ 0.01 and IC$_{50}$ ≤ 50 nM). The designation of a polypeptide as immunogenic was based on its content of T-cell peptides with affinity for MHC class II molecules. Accordingly, we also report the number of T-cell peptides per polypeptide. For the WB isolate, 1,402 T-cell peptides were identified (100%), of which 34% showed affinity for I-E$^k$, 31% for I-A$^b$, 21% for I-A$^d$, and 14% for I-E$^d$ (Figure C). For the

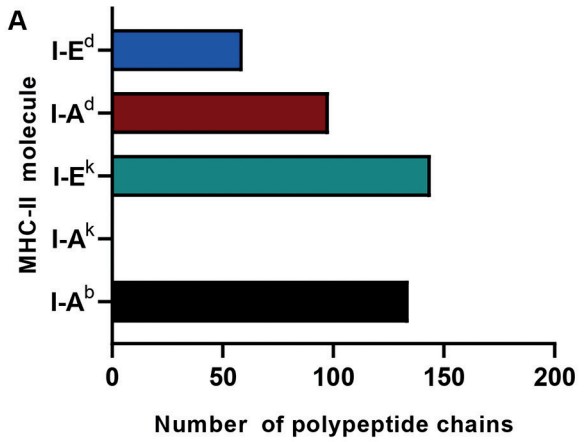

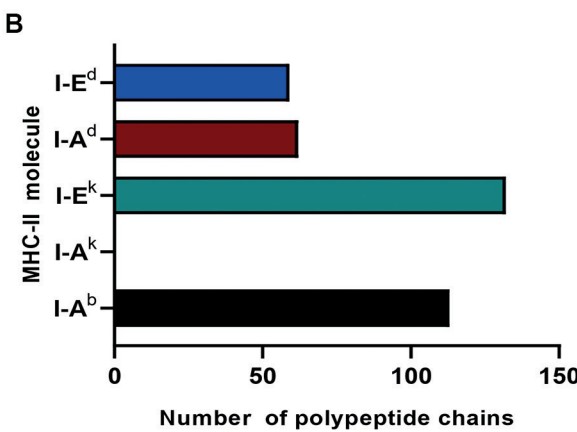

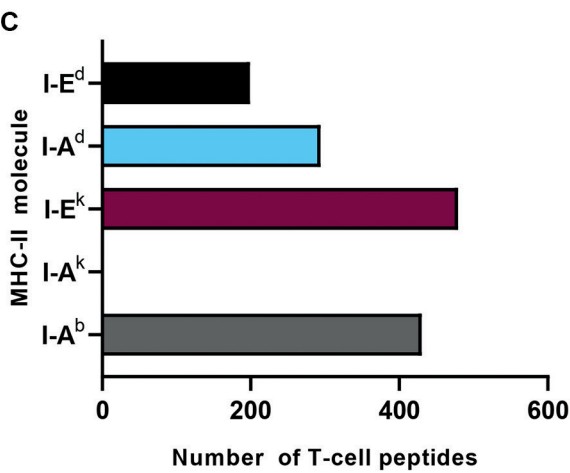

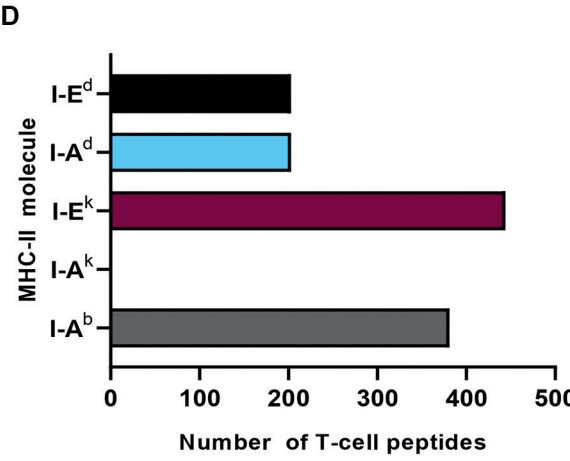

The *Giardia lamblia* proteome contains immunogenic polypeptides and T-cell peptides that predominantly interact with I-A$^b$ and I-E$^k$ molecules. The *Giardia* proteome was obtained from GiardiaDB and analysed using NetMHCII 2.3 via immune epitope database (IEDB). A Python 3.9 script was developed to process the data. The analysis included five murine major histocompatibility complex (MHC) class II molecules: I-E$^d$, I-A$^d$, I-E$^k$, I-A$^k$, and I-A$^b$. The graphics show the distribution of polypeptides and T-cell peptides for each MHC class II molecule. Panels A and C correspond to *Giardia* assemblage A isolate WB, while panels B and D correspond to *Giardia* assemblage B isolate GS.

GS isolate, 1,231 T-cell peptides were identified (100%), with 36% showing affinity for I-E$^k$, 31% for I-A$^b$, and 16% each for I-A$^d$ and I-E$^d$ (Figure D). No peptides with affinity for I-A$^k$ were identified under the selected parameters for either isolate.

We also performed a homology analysis to identify immunogenic polypeptides with similar amino acid sequences between the WB and GS isolates [Supplementary data (Table I)]. The amino acid sequences of the 414 polypeptide chains identified from the WB isolate were compared with the 350 chains from the GS isolate. A total of 129 polypeptides exhibited ≥ 80% sequence homology. The T-cell peptides from each polypeptide are presented, highlighting that each pair of homologous polypeptide chains shares the same peptide core.

Antigen presentation can be enhanced when peptides are capable of binding to more than one MHC molecule. Our results showed that 21 polypeptide chains out of 414 in the WB isolate contained T-cell peptides that could be presented by multiple MHC class II molecules [Supplementary data (Table II)]. These molecules are referred to as promiscuous polypeptide chains. Sixteen promiscuous polypeptide chains were also identified in the GS isolate [Supplementary data (Table III)].

*The immunogenic proteins of the secretome of G. lamblia - G. lamblia* is unable to cross the intestinal epithelial barrier, which raises questions about how this protozoan triggers an adaptive immune response. One possible explanation involves the excretory-secretory products of *G. lamblia*, which constitute the *Giardia* secretome.

We therefore investigated whether any of the identified immunogenic polypeptides were also present in the *Giardia* secretome. A comparative analysis was performed between the immunogenic polypeptides identified in this study and the molecules detected in the secretomes of *G. lamblia* isolates WB and GS.[9] For the WB isolate, 12 immunogenic polypeptides (3% of 414) were found to belong to the secretome (Table I). These antigens include cathepsins, high-cysteine membrane proteins, and enzymes involved in glycolysis. The T-cell peptides generated from each polypeptide are also listed, with the peptide core sequence shown; the core is identical for all peptides derived from the same polypeptide.

For the GS isolate, 10 polypeptides (3% of 350) were found to be part of the secretome (Table II). This group includes proteins such as cathepsin B, high-cysteine membrane proteins, and enzymes involved in the glycolysis pathway. As with the WB isolate, the peptide core sequence of the T-cell peptides is identical for all peptides derived from the same polypeptide.

Notably, all immunogenic proteins identified in the secretomes of both *G. lamblia* isolates showed affinity for I-A$^b$. Additionally, only one protein from each isolate exhibited affinity for the I-A$^d$ class II molecule.

*Specific interactions of the immunogenic T-cell peptides of G. lamblia* - MHC class II molecules consist of two polymorphic chains, α and β, which together form the peptide-binding groove. This groove interacts with a sequence of nine amino acids, known as the peptide core. The groove contains pockets that bind specific amino acids within the peptide core, thereby influencing the affinity of the MHC class II molecule for a particular T-cell peptide.

The anchor residues for the I-A$^d$ molecule are located in pockets P1, P4, and P9, with the model antigen corresponding to the OVA$_{323-339}$ peptide (SQAVHAAHA). Pocket P1 is the least restrictive, accommodating amino acids of various sizes and charges with minimal impact on peptide-binding affinity. Pocket P4 is the most restrictive, preferentially accommodating small, uncharged amino acids. Pocket P9 is the second most restrictive and also favours small, uncharged amino acids. As shown in Table III, T-cell peptides from both isolates contain methionine in pocket P4 and alanine in pocket P9. For pocket P1, peptide 114-128 (VIKMSALPA) from the WB isolate contains valine, whereas peptide 194–208 (LKAMKAVAA) from the GS isolate contains leucine.

The I-E$^k$ molecule has a peptide-binding groove with anchor residues located in pockets P1, P4, P6, and P9. Pockets P1 and P9 are the most restrictive. Pocket P1 accommodates small hydrophobic amino acids such as valine, leucine, and isoleucine, while pocket P9 is suited to large hydrophilic amino acids, preferentially lysine or arginine. Hydrophobic amino acids predominate in pocket P4, whereas charged amino acids are common in pocket P6. As shown in Table III, T-cell peptides from both isolates, GS$_{375-389}$ (VIRMIYFYK) and WB$_{373-387}$ (VIRMIYFYK), contain valine at P1 and lysine at P9, consistent with the model antigen Hsp70$_{236-248}$ (VNHFIAEFK).

The peptide-binding groove of the I-A$^k$ molecule features anchor residues in pockets P1, P4, P6, and P9. Pocket P1 is the most restrictive, accommodating negatively charged amino acids, with a preference for aspartic acid. Pocket P6 is the second most restrictive, preferentially accommodating glutamic acid and glutamine. Although pocket P4 is less restrictive, it preferentially accommodates medium-sized hydrophobic amino acids such as isoleucine, valine, and leucine. Pocket P9 is the least restrictive, allowing a variety of residues to fit.

As shown in Table III, the model antigen peptide (hen egg white lysozyme, HEL$_{50-62}$, DYGILQINS) presents aspartic acid in pocket P1, whereas the peptide from the WB isolate contains histidine and that from the GS isolate contains asparagine. For pocket P4, HEL exhibits isoleucine, while the T-cell peptides from the WB and GS isolates present tyrosine and serine, respectively. The model antigen shows glutamine in pocket P6, as does the peptide from WB, whereas the peptide from GS contains histidine. Finally, for pocket P9, the peptide from WB contains isoleucine, whereas the peptide from GS contains histidine.

## DISCUSSION

In the present study, we performed an immunoinformatic analysis of the entire proteome of *G. lamblia* assemblages A and B to identify potential immunogenic antigens. Notably, in both assemblages, most polypeptides exhibited higher affinity for I-A$^b$ and I-E$^k$ molecules, emphasising the influence of the structural characteristics of each MHC class II binding groove.

TABLE I

Immunogenic polypeptides present in the secretome of *Giardia* assemblage A isolate WB

| Polypeptide ID | Polypeptide name | TM domains | Polypeptide length | Molecular weight (kDa) | T-cell peptides | MHC-II |
|---|---|---|---|---|---|---|
| GL50803_006330 | Unspecified product | No | 271 | 29 | $_{194}$KWYSYHAALATDLLL$_{208}$<br>$_{192}$ATKWYSYHAALATDL$_{206}$<br>$_{193}$TKWYSYHAALATDLL$_{207}$ | I-A$^b$ |
| GL50803_0016779[1] | Cathepsin B | No | 298 | 33 | $_6$LAAAAFSAPALTVSE$_{20}$, $_5$LLAAAAFSAPALTVS$_{19}$<br>$_7$AAAAFSAPALTVSEL$_{21}$, $_8$AAAFSAPALTVSELN$_{22}$<br>$_3$LFLLAAAAFSAPALT$_{17}$, $_4$FLLAAAAFSAPALTV$_{18}$ | I-A$^b$ |
| GL50803_0014019[4] | Cathepsin B | No | 300 | 33 | $_8$AAAFSAPALTVSELN$_{22}$, $_3$LFLLAAAAFSAPALT$_{17}$<br>$_4$FLLAAAAFSAPALTV$_{18}$, $_7$AAAAFSAPALTVSEL$_{21}$<br>$_6$LAAAAFSAPALTVSE$_{20}$, $_5$LLAAAAFSAPALTVS$_{19}$ | I-A$^b$ |
| GL50803_002834 | Putative GTPase activating protein for ARF | No | 398 | 42 | $_{211}$QTAPAMFTAPAVSTM$_{225}$<br>$_{213}$APAMFTAPAVSTMPA$_{227}$<br>$_{212}$TAPAMFTAPAVSTMP$_{226}$ | I-A$^b$ |
| GL50803_009117[2] | cAMP-dependent protein kinase regulatory chain | No | 460 | 51 | $_{46}$IHKFASYSPLAAAVL$_{60}$ | I-A$^b$ |
| GL50803_008528 | Unspecified product | No | 528 | 57 | $_{236}$FDVLVTFQPAMAAKL$_{250}$ | I-A$^b$ |
| GL50803_0027717 | High cysteine membrane protein Group 3 | Yes | 836 | 86 | $_{245}$GRCISAFSAASALAG$_{259}$<br>$_{246}$RCISAFSAASALAGC$_{260}$<br>$_{247}$CISAFSAASALAGCS$_{261}$, $_{248}$ISAFSAASALAGCST$_{262}$<br>$_{249}$SAFSAASALAGCSTY$_{263}$ | I-A$^b$ |
| GL50803_0013922 | Unspecified product | Yes | 1087 | 114 | $_{238}$VFRYSVASAALVKGK$_{252}$<br>$_{234}$YQNAVFRYSVASAAL$_{248}$<br>$_{237}$AVFRYSVASAALVKG$_{251}$<br>$_{235}$QNAVFRYSVASAALV$_{249}$<br>$_{236}$NAVFRYSVASAALVK$_{250}$ | I-A$^b$ |
| GL50803_00102101 | Kinesin-3 | No | 1026 | 115 | $_{728}$KFLRKYSALRALFSH$_{742}$ | I-A$^d$ |
| GL50803_0016125[3] | FAD-dependent glycerol-3-phosphate dehydrogenase | No | 1111 | 119 | $_{358}$IMYSFASARAVLPSN$_{372}$,<br>$_{359}$MYSFASARAVLPSND$_{373}$<br>$_{354}$VPEEIMYSFASARAV$_{368}$<br>$_{357}$EIMYSFASARAVLPS$_{361}$<br>$_{355}$PEEIMYSFASARAVL$_{369}$<br>$_{356}$EEIMYSFASARAVLP$_{370}$ | I-A$^b$ |
| GL50803_0015591 | Coiled-coil protein | No | 1595 | 176 | $_{111}$ILSLYASAALASAVA$_{125}$ | I-A$^b$ |
| GL50803_0017476 | High cysteine membrane protein | Yes | 2169 | 224 | $_{451}$VALFYTYTAANAVSS$_{465}$<br>$_{452}$ALFYTYTAANAVSSY$_{466}$<br>$_{449}$SKVALFYTYTAANAV$_{463}$<br>$_{454}$FYTYTAANAVSSYII$_{468}$<br>$_{450}$KVALFYTYTAANAVS$_{464}$<br>$_{453}$LFYTYTAANAVSSYI$_{467}$ | I-A$^b$ |

Peptide core is highlighted in grey. The polypeptide ID of conserved polypeptides between isolate WB and GS is red coloured. The pair of conserved polypeptides is denoted by a superscript number. *Homology with polypeptides of isolate WB does not present in the secretome. TM: transmembrane domains; MHC-II: major histocompatibility complex class II.

The results shown in Figure indicate that immunogenic polypeptides bind to MHC class II molecules in different ways. The I-E$^k$ allele exhibited higher affinity for most polypeptides, whereas the I-A$^k$ allele did not show high affinity for any of the polypeptides analysed. This is likely due to the distinct structural requirements of the binding groove of each MHC class II molecule. The I-A$^k$ molecule possesses one of the most restrictive binding grooves, with the principal constraint being the requirement for peptides to contain a negatively charged residue — primarily aspartic acid — at the P1 position of the core, in order to form stable, long-lasting interactions with the I-A$^k$ molecule.[21,23]

This effect is illustrated in Table III, which shows the interactions of T-cell peptides with the highest affinity for MHC class II molecules (I-A$^d$, I-E$^k$, and I-A$^k$) from both WB and GS isolates. The low affinity of peptides for I-A$^k$ molecules may be due to their inability to fit

TABLE II

Immunogenic polypeptides present in the secretome of *Giardia* assemblage B isolate GS

| Polypeptide ID | Polypeptide name | TM domains | Polypeptide length | Molecular weight (kDa) | T-cell peptides | MHC-II |
|---|---|---|---|---|---|---|
| GL50581_4499 | Hypothetical protein | No | 281 | 30 | $_{76}$GVLV**FKAAEPVASED**$_{90}$, $_{74}$SD**GVLVFKAAEPVAS**$_{88}$ <br> $_{75}$D**GVLVFKAAEPVASE**$_{89}$, $_{77}$**VLVFKAAEPVASEDE**$_{91}$ | I-A$^b$ |
| GL50581_78$^{1,4}$ | Pept_C1 domain-containing protein | No | 298 | 33 | $_{4}$F**LLAAAAFSAPALTV**$_{18}$, $_{3}$**LFLLAAAAFSAPALT**$_{17}$ <br> $_{8}$**AAAFSAPALTVSELN**$_{22}$, $_{7}$**AAAAFSAPALTVSEL**$_{21}$ <br> $_{6}$**LAAAAFSAPALTVSE**$_{20}$, $_{5}$**LLAAAAFSAPALTVS**$_{19}$ | I-A$^b$ |
| GL50581_1446$^{2}$ | cAMP-dependent protein kinase regulatory chain | No | 460 | 50 | $_{46}$IHK**FASYSPLAAAVL**$_{60}$ | I-A$^b$ |
| GL50581_3080* | M20_dimer domain-containing protein | No | 506 | 55 | $_{393}$**VYESLKALASLAGFS**$_{407}$ | I-A$^d$ |
| GL50581_1032* | ANK_REP_REGION domain-containing protein | No | 623 | 66 | $_{377}$**AAIHAANAAAATAND**$_{391}$ <br> $_{376}$**NAAIHAANAAAATAN**$_{390}$ <br> $_{378}$**AIHAANAAAATANDN**$_{392}$ <br> $_{375}$**ANAAIHAANAAAATA**$_{389}$ | I-A$^b$ |
| GL50581_3032 | Kinase, NEK | No | 782 | 86 | $_{517}$PVEP**VYASAPVALAE**$_{531}$, $_{518}$**VEPVYASAPVALAEL**$_{532}$ | I-A$^b$ |
| GL50581_411 | High cysteine membrane protein Group 2 | Yes | 834 | 89 | $_{711}$**AIDGYYYNSAKASVT**$_{725}$ <br> $_{713}$**DGYYYNSAKASVTQC**$_{727}$ <br> $_{712}$**IDGYYYNSAKASVTQ**$_{726}$ | I-A$^b$ |
| GL50581_1272* | Protein 21.1 | No | 1032 | 113 | $_{663}$**ETRAFAAACVNQMQE**$_{677}$ <br> $_{661}$**LEETRAFAAACVNQM**$_{675}$ <br> $_{662}$**EETRAFAAACVNQMQ**$_{676}$ | I-A$^b$ |
| GL50581_2252$^{3}$ | Glycerol-3-phosphate dehydrogenase | No | 1111 | 119 | $_{356}$**KEIMYSFASARAVLP**$_{370}$ <br> $_{357}$**EIMYSFASARAVLPC**$_{371}$ <br> $_{355}$**PKEIMYSFASARAVL**$_{369}$ <br> $_{358}$**IMYSFASARAVLPCN**$_{372}$ <br> $_{354}$**TPKEIMYSFASARAV**$_{368}$ <br> $_{359}$**MYSFASARAVLPCND**$_{373}$ | I-A$^b$ |
| GL50581_881 | Hypothetical protein | No | 1904 | 214 | $_{1291}$**YIRVYMGSTAAPAAA**$_{1305}$ <br> $_{1292}$**IRVYMGSTAAPAAAP**$_{1306}$ | I-A$^b$ |

Peptide core is highlighted in grey. The polypeptide ID of conserved polypeptides between isolate WB and GS is red coloured. The pair of conserved polypeptides is denoted by a superscript number. *Homology with polypeptides of isolate WB does not present in the secretome. TM: transmembrane domains; MHC-II: major histocompatibility complex class II.

properly into the binding groove, resulting from the absence of negatively charged amino acids, such as aspartic acid or glutamic acid.

The I-E$^k$ allele was the MHC class II molecule that bound the greatest number of polypeptides and T-cell peptides from both isolates (Figure A and C). Several studies report that most of the peptides eluted from I-E$^k$ contain hydrophobic amino acids — primarily valine, leucine, and isoleucine — at pocket P1. Nevertheless, other residues with hydrophobic properties can also occupy this pocket.[24,25,26] Most polypeptides from both isolates contain T-cell peptides with hydrophobic amino acids at the P1 position, which may explain why a larger number of polypeptides exhibited higher affinity for I-E$^k$ compared with the other MHC class II molecules analysed in this study.

An ideal vaccine antigen should confer cross-protection against different strains from the two genetic assemblages of *G. lamblia* that cause human disease. In this study, we analysed conserved immunogenic polypeptides between assemblages A and B and identified several highly conserved polypeptides with greater than 80% homology [Supplementary data (Table I)]. Among these, some proteins — such as dyneins and NEK kinase — are considered constitutive, as they are expressed throughout all stages of the parasite's cell cycle.

Additionally, several of the identified polypeptides correspond to molecules that perform key biological functions in *Giardia* or act as virulence factors. Some of these conserved polypeptides belong to the family of proteases known as cathepsins, with cathepsin B-like proteases being the most highly expressed in *Giardia*.[27]

TABLE III

Interactions of immunogenic T-cell peptides of *Giardia* isolates WB and GS with the binding groove of MHC-II molecules

| | | Peptide core | | | | | | | | | |
|---|---|---|---|---|---|---|---|---|---|---|---|
| | | P1 | P2 | P3 | P4 | P5 | P6 | P7 | P8 | P9 | IC$_{50}$ (nM) |
| I-A$^d$ | OVA (323-339) | S | Q | A | V | H | A | A | H | A | |
| | WB (114-128) | V | I | K | M | S | A | L | P | A | 5.2 |
| | GS (194-208) | L | K | A | M | K | A | V | A | A | 6.5 |
| I-E$^k$ | Hsp70 (236-248) | V | N | H | F | I | A | E | F | K | |
| | WB (375-389) | V | I | R | M | I | Y | F | Y | K | 7.8 |
| | GS (373-387) | V | I | R | M | I | Y | F | Y | K | 7.8 |
| I-A$^k$ | HEL (50-62) | D | Y | G | I | L | Q | I | N | S | |
| | WB (204-218) | H | Q | D | Y | N | Q | N | Q | I | 523.2 |
| | GS (125-139) | N | A | H | S | A | H | S | N | H | 566.6 |

MHC-II: major histocompatibility complex class II.

Only a few complete epitopes were found to be conserved among the polypeptides of the assemblages analysed in this study. However, we identified numerous peptide cores that were highly conserved within the corresponding immunogenic polypeptides of *Giardia*. These highly conserved peptide cores appear to be minimally affected by parasite adaptation and immune evasion, or they may form part of functionally important domains of the polypeptides.

Promiscuity can enhance the host immune response by increasing the likelihood that a polypeptide will be presented to T cells. In this study, we defined promiscuous polypeptides as those generating T-cell peptides capable of binding to more than one MHC class II molecule. We identified 21 promiscuous polypeptides in the WB isolate [Supplementary data (Table II)] that can bind to different MHC class II molecules, whereas 16 promiscuous polypeptides were found in the GS isolate [Supplementary data (Table III)]. Although MHC molecules are highly polymorphic, H2 shares key conserved regions with human HLA antigens.[28] It would be of interest to investigate whether these polypeptides can be recognised by one or more human HLA molecules.

Previous studies have reported that individuals infected with *Giardia* express the HLA haplotypes HLA-DRB103:01, HLA-DRB113:01, and HLA-DRB107:01. In contrast, individuals expressing HLA-DRB104:02, *HLA-DRB1*10:01, HLA-DRB114:01, and HLA-DRB1*15:01 appear to be resistant to the parasite.[6,29,30] This is particularly noteworthy because the peptide-binding groove of I-E$^k$ has been reported to be highly similar to that of HLA-DR molecules.[20]

*Giardia lamblia* is a luminal parasite, meaning that it cannot cross the epithelial barrier. This raises questions regarding the mechanisms employed by the protozoan to elicit an immune response. The study of the excretory-secretory products (ESPs) of *G. lamblia*, collectively referred to as the *Giardia* secretome,[31] may help to elucidate how this parasite stimulates host immune responses. Among the various secreted molecules, cysteine proteases (CPs) are recognised as major virulence factors and play a critical role during infection.

Our results indicate that this group of proteins is highly immunogenic and is present in the secretome of both isolates (Tables I-II). In addition to previous studies emphasising their role in virulence during infection, these proteins may serve as potential targets for the development of anti-*Giardia* therapies[32] or as tools to improve our understanding of the parasite's behaviour and its interaction with the host immune system.

High-cysteine membrane proteins (HCMPs) have also been identified as components of *Giardia* ESPs and represent the most abundant differentially expressed genes when *Giardia* interacts with intestinal epithelial cells.[33,34]

Assemblages A and B are responsible for human giardiasis; therefore, antigens conserved across both assemblages, as well as those present in the *Giardia* secretome, represent ideal vaccine candidates. In this study, we identified all potential immunogenic antigens of this parasite using immunoinformatics. Among these, six polypeptides were conserved in both assemblages and present in the secretome: cathepsin B (GL50803_0016779), protein kinase regulatory chain (GL50803_009117), glycerol-3-phosphate dehydroge-

nase (GL50803_0016125), M_20 dimer domain-containing protein (GL50581_3080), ANK_REP_REGION domain-containing protein (GL50581_1032), and protein 21.1 (GL50581_1272). Two to three T-cell peptides from each of these polypeptides could be incorporated into a multiepitope vaccine against *Giardia*. Multiepitope vaccine technology has been widely applied in the design and development of vaccines against a range of pathogens, including dengue virus, HIV-1, influenza virus, and *Leishmania*,[35,36,37,38] making it a promising strategy for giardiasis vaccine design. While GiardiaDB contains some unvalidated entries, which should be considered when selecting candidate antigens, the selected candidates display the key features of immunogenic proteins, emphasising their strong potential for inclusion in a vaccine.

In the present study, we conducted an immunoinformatic analysis to identify immunogenic polypeptides in *G. lamblia* assemblages A and B. While immunoinformatic tools are invaluable for integrating large-scale immunological data in a short period, thereby facilitating drug and vaccine development, several limitations remain to be considered. Based on our final cut-off criteria, we did not identify polypeptides corresponding to proteins previously described as immunogenic, such as CWPs, giardins, and tubulins. Nevertheless, initial analyses using less stringent thresholds indicated that several polypeptides from CWPs, VSPs, giardins, and metabolic proteins contained MHC class II-restricted peptides with percentile ranks of 0.1% or 1%, which are still considered strong binders. These proteins have been experimentally confirmed as immunogenic.[6] Furthermore, we evaluated the selected polypeptides across multiple threshold combinations [Supplementary data (Tables IV-V)]. Collectively, these findings underscore the robustness and feasibility of our approach for identifying immunogenic candidates. Additionally, our analysis revealed that none of the *Giardia* polypeptides were predicted to be immunogenic for I-A$^k$ under the parameters selected, despite experimental evidence from our research group demonstrating the immunogenic activity of *Giardia* antigens for I-A$^k$.[39,40,41] Although experimentally validated *Giardia* T-cell epitopes are not yet available, our predictions provide a valuable foundation for future laboratory validation, thereby advancing the identification of potential vaccine targets. These findings emphasise the limitations of immunoinformatic tools in capturing all aspects of immunogenicity during rapid, large-scale epitope screening. For MHC class II prediction servers, it would be advantageous to additionally consider factors such as antigen processing, the flanking amino acid sequences, interactions with the T-cell receptor, and the potential effector mechanisms of T cells. Moreover, during the processing of exogenous antigens, HLA-DM accessory molecules play a critical role in the selection of immunodominant epitopes and their restricted presentation by MHC class II molecules.[42,43] Despite advances in T-cell epitope prediction through *in silico* approaches, there remains a need for improved tools capable of accurately identifying immunogenic MHC class II epitopes. Such tools should incorporate additional determinant features — such as those described above—to enhance the precision of epitope screening. The limitations outlined previously underscore the importance of validating MHC class II predictions *in vivo* to confirm the true immunogenicity of the selected polypeptides. In this context, the value of performing MHC class II predictions using murine alleles becomes apparent. This strategy allows for *in vivo* testing of predicted immunogenicity, as the selected murine MHC class II alleles correspond to some of the most commonly used mouse strains in *Giardia* research: C57BL/6 (haplotype b), BALB/c (haplotype d), and C3H/HeJ (haplotype k). With regard to experimental validation, the ideal next step would be to select a subset of polypeptides and perform analyses in one or more of the murine models mentioned above to confirm immunogenicity. The resulting data will inform the potential use of these polypeptides in future studies involving human models.

Another limitation of our study is the reliance on GiardiaDB as the source of *Giardia* proteomes. Like other biological databases (*e.g.*, GenBank, IEDB, UniProt), GiardiaDB contains protein sequences that are computational predictions and may include redundancies or unvalidated entries. Consequently, some of the immunogenic polypeptides identified may not correspond to bona fide *Giardia* proteins.

To our knowledge, this is the first study to analyse the complete proteome of *Giardia* with the aim of identifying all its potential immunogenic antigens. This knowledge will contribute to the future development of novel prophylactic approaches against giardiasis and pave the way for implementing a combinatorial strategy that integrates immunoinformatics and data science to investigate other pathogens beyond *G. lamblia*.

## AUTHORS' CONTRIBUTION

DOT - writing-original draft, methodology, conceptualisation, formal analysis; CAVV - software, formal analysis; TG, GLR and LBP - writing-review & editing; CV writing-review & editing, conceptualisation, funding acquisition, methodology and formal analysis. The authors declare no conflicts of interest.

## DATA AVAILABILITY

The data that support the findings of this study are openly available in GitHub at https://github.com/davidortega47-rgb/Datasets_Immunoinformatic_Prediction_G.lamblia.git and https://github.com/davidortega47-rgb/LIBCE-codes.git. Also, these data can be found in Zenodo at https://doi.org/10.5281/zenodo.18210685 and https://doi.org/10.5281/zenodo.18166880. Supplementary data associated with this article can be found in the online version.

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

# OPEN PEER REVIEW

Memórias do IOC thanks the anonymous reviewers for their contribution to the peer review of this work.

## FIRST REVIEW ROUND

REVIEWERS' COMMENTS

### REVIEWER #1

The abstract clearly states the rationale (lack of human vaccine), the aim (systematic immunoinformatic screen), methodology (NetMHCII 2.3, murine MHC-II alleles, Python pipeline), main results (414 immunogenic WB, 350 GS, conserved/promiscuous polypeptides, secretome hits), and significance (first comprehensive proteome-wide map).

It is concise, well-structured, and understandable for a broad readership. However, it is written in a very technical language: IC50 values and percentile thresholds may be too detailed for the abstract.

The clinical relevance is mentioned only in the last line; could be emphasized earlier. "Data science" is mentioned vaguely. It remains unclear what beyond the NetMHCII predictions was novel. Taken together, the structure is fine but the abstract would benefit from reducing technical jargon, and highlighting the potential translational impact more clearly.

The study is original and the claim to be the first complete proteome-wide immunoinformatic screen for Giardia is valid and novel. It provides a foundational catalog of candidate immunogens, which is highly relevant given the absence of a licensed vaccine and the burden of giardiasis globally. However, the translational step from murine MHC-II predictions to human HLA applicability is somewhat underdeveloped, although parallels with HLA-DR are discussed. Most importantly, the fact that experimental validation is absent limits the impact of this work. Hence, conclusions about vaccine candidacy should be toned down.

The authors rely on a single data source (GiardiaDB proteomes, secretome datasets) which is appropriate but requires explicit consideration of limitations. While the genome assemblies in GiardiaDb are robust, automated annotation and gene model prediction is not validated. It has to be made clear in the text that protein sequences are unvalidated predictions and only as good as the gene models are.

The use of NetMHCII 2.3 and stringent cutoffs (≤0.01 percentile, IC50 ≤ 50 nM) provides rigor, though may be overly restrictive, missing known epitopes.

Homology and secretome integration are well described.

A limitation is that only murine alleles were tested. The lack of direct HLA predictions weakens translational relevance.

The results are comprehensive, with clear reporting of numbers of immunogenic proteins, conserved homologs, secretome overlaps, and promiscuous peptides.

There are too many tables and figures. I recommend organizing and moving this data to deposit in an open access repository and make the links available in a newly created Gitlab repository for the project.

Albeit strong in contextualizing findings with immunobiology of Giardia, MHC binding properties, and parallels to human immunity, the discussion is exceedingly long and needs to be reorganized to serve its direct purpose and significantly shortened. The authors should appropriately acknowledge the limitations of prediction tools, the need for in vivo validation, and the missed known epitopes.

AUTHORS' RESPONSE TO THE REVIEWERS

Hermosillo, Sonora, November 10th, 2025
Dr. Ernesto Caffarena
Handling Editor
Memórias do Instituto Oswaldo Cruz

Dear Dr. Caffarena,

Enclosed please find the revised version of our manuscript entitled "Immunoinformatic analysis for identifying immunogenic antigens from the complete proteome of Giardia lamblia," previously submitted to Memórias do Instituto Oswaldo Cruz under ID MIOC-2025-0216.

The manuscript has been carefully revised according to the reviewers' comments, and this document provides a detailed, point-by-point response to each of their observations. We remain at your disposal for any further clarification or additional information you may require.

Sincerely,
Dr. Carlos Velazquez
Department of Chemistry-Biology
Universidad de Sonora

Responses to reviewers
Reviewer 1
Reviewer 1 required attention about the following aspects:

Comment 1: "It is concise, well-structured, and understandable for a broad readership. However, it is written in a very technical language: IC50 values and percentile thresholds may be too detailed for the abstract" and "The clinical relevance is mentioned only in the last line; could be emphasized earlier. "Data science" is mentioned vaguely. It remains unclear what beyond the NetMHCII predictions was novel. Taken together, the structure is fine but the abstract would benefit from reducing technical jargon and highlighting the potential translational impact more clearly".

Answer 1: We thank the reviewer for these thoughtful comments and agree with the suggestions. The clinical relevance of this work is now mentioned earlier in the abstract (page 2, lines 29-32), and the potential translational applications are described more clearly (page 2, lines 44-46). Although the terms IC50 and percentile rank are standard cut-off criteria for selecting strong-binding peptides, they are already described in the main text and have therefore been removed from the abstract. The revised abstract now refers more generally to "strong binder peptides" or "sequences with high affinity for MHC class II molecules". In addition, we now clarify the distinction between immunoinformatics and data science in the Materials and Methods section: immunoinformatics refers to the use of epitope prediction and promiscuity tools, whereas data science refers to the use of Python 3.9 to develop a pipeline for processing and analyzing large datasets obtained from the IEDB and GiardiaDB databases. This approach allowed us to integrate and automate analyses beyond standard NetMHCII predictions, highlighting the novel computational contribution of our study.

Comment 2: "The study is original and the claim to be the first complete proteome-wide immunoinformatic screen for Giardia is valid and novel. It provides a foundational catalog of candidate immunogens, which is highly relevant given the absence of a licensed vaccine and the burden of giardiasis globally. However, the translational step from murine MHC-II predictions to human HLA applicability is somewhat underdeveloped, although parallels with HLA-DR are discussed. Most importantly, the fact that experimental validation is absent limits the impact of this work. Hence, conclusions about vaccine candidacy should be toned down"

Answer 2: We appreciate the reviewer's thoughtful feedback and concur with the proposed suggestions. The manuscript has been revised to address the limitations of experimental models concerning the translation of vaccine candidates to humans (page 16, lines 371-375). The murine model is widely used in Giardia immunology research and provides a valuable starting point for identifying novel candidates that may confer protection against infection. Additionally, predictions based on murine MHC-II alleles can guide future studies exploring human HLA applicability. Regarding experimental validation, the ideal next step would be to select a subset of polypeptides and perform experimental analyses to confirm their immunogenicity. We have accordingly tempered the language in the conclusions to reflect these limitations and avoid overstating vaccine candidacy.

Comment 3: "The authors rely on a single data source (GiardiaDB proteomes, secretome datasets) which is appropriate but requires explicit consideration of limitations. While the genome assemblies in GiardiaDb are robust, automated annotation and gene model prediction is not validated. It has to be made clear in the text that protein sequences are unvalidated predictions and only as good as the gene models are¨

Answer 3: We thank the reviewer for this important comment and agree with the observation. GiardiaDB (https://giardiadb.org/giardiadb/app) is a repository containing genome and proteome data from several Giardia genotypes. Some data have been validated using microarrays and mass spectrometry, and deprecated genes have been discarded (Aurrecoechea C., 2008). However, we acknowledge that, like other biological databases (e.g., GenBank, IEDB, UniProt), GiardiaDB contains protein sequences that are computational predictions and may include redundancies or unvalidated entries. These limitations have now been explicitly acknowledged in the manuscript (pages 16-17, lines 376-380). Despite these limitations, the dataset provides a comprehensive starting point for identifying potential immunogens and guiding future experimental validation.

Comment 4: "The use of NetMHCII 2.3 and stringent cutoffs (≤0.01 percentile, IC50 ≤ 50 nM) provides rigor, though may be overly restrictive, missing known epitopes"

Answer 4: We appreciate the reviewer's valuable feedback and concur with the observation. Commonly used affinity-based thresholds for T cell epitopes are IC50 < 500 nM and percentile rank < 10% (Reardon B., 2021; Paul S., 2013), corresponding to approximately 80% sensitivity in predicting true ligands. By applying more stringent cut-off points (IC50 < 50 nM and rank < 0.01), we aimed to select the most immunogenic antigens from the ~4,500 proteins comprising the Giardia proteome. We initially evaluated less stringent cut-off points (rank < 0.1, rank < 1, rank < 5), but these yielded over 1,000 polypeptides per genotype, which was impractical for downstream analyses. We acknowledge that these stringent thresholds may exclude some epitopes, constituting a limitation of the study, which is discussed in the manuscript (page 15, lines 347-357). Nevertheless, this approach ensures that the selected candidates are highly likely to be immunogenic, providing a focused set of targets for future experimental validation.

Comment 5: "There are too many tables and figures. I recommend organizing and moving this data to deposit in an open access repository and make the links available in a newly created Gitlab repository for the project"

Answer 5: We thank the reviewer for the helpful feedback and agree with the proposed suggestion. Some tables have been moved to the Supplementary Material section of the revised manuscript.

Comment 6: "The discussion is exceedingly long and needs to be reorganized to serve its direct purpose and significantly shortened. The authors should appropriately acknowledge the limitations of prediction tools, the need for in vivo validation, and the missed known epitopes"

Answer 6: We are grateful for the reviewer's thoughtful remarks and accept the suggestions made. General topics, such as Giardia immunobiology and MHC properties, have been abbreviated to streamline the Discussion. The limitations of using prediction tools, including the potential to miss known epitopes and the necessity for in vivo validation, have been explicitly acknowledged in the revised Discussion section (pages 15-16, lines 344-375).

## SECOND REVIEW ROUND

### REVIEWERS' COMMENTS

**REVIEWER #1**

Manuscript ID MIOC-2025-0216.R1 entitled "Immunoinformatic analysis for identifying immunogenic antigens from the complete proteome of Giardia lamblia" which you submitted to the Memórias do Instituto Oswaldo Cruz, has been fully evaluated again, and some concerns were raised about the manuscript as it currently stands.

These issues must be addressed before we would be willing to consider a revised version of your study.

Since the original reviewer is unavailable, I have personally assumed responsibility for this assessment.

After thorough examination, I must inform you that the manuscript still requires substantial revisions before it can be reconsidered for publication. While I acknowledge your efforts to address several of the reviewer's concerns, three critical issues require substantial additional work before acceptance.

First and most critically, you have completely failed to address the reviewer's Comment 5 regarding open data accessibility. The reviewer made a specific recommendation to deposit your data in an open access repository and create a public code repository. You merely moved some tables to Supplementary Material, which does not constitute open data sharing. This represents a fundamental failure to comply with Open Science principles that are non-negotiable for publication in this journal. You must create a public repository (GitHub, GitLab, or equivalent) containing all analysis code, peptide prediction datasets, and processing scripts. Additionally, deposit your complete datasets in a recognized repository such as Zenodo, Figshare, or Dryad with a DOI. Include all repository URLs and DOI numbers in your Data Availability Statement. Without this, the manuscript cannot be accepted.

Second, the reviewer raised concerns about your overly restrictive cut-off thresholds potentially missing relevant epitopes. Your justification based on practical convenience is scientifically insufficient. You must perform a sensitivity analysis by re-running your predictions using at least two additional thresholds (e.g., IC50 <100 nM and <500 nM, or percentile rank <0.1 and <1.0). Present a summary table in the supplementary material showing: (a) the total number of candidate polypeptides identified at each threshold, (b) the number of these showing promiscuous binding, and (c) overlap between threshold groups. If experimentally validated Giardia lamblia T-cell epitopes are available in the published literature, demonstrate whether your thresholds capture them; if no such epitopes exist, explicitly state this limitation. This analysis will allow assessment of the robustness of your candidate selection strategy.

Third, regarding Comment 3, the reviewer explicitly stated that it must be made clear throughout the text that GiardiaDB protein sequences are unvalidated computational predictions. Five lines buried in the Discussion is insufficient for a limitation that affects the validity of your entire dataset. You must add explicit statements in the Introduction when first presenting GiardiaDB as your data source, in the Methods section when describing protein sequence retrieval, and in your Conclusions when presenting candidate antigens. This level of transparency about data quality is fundamental to scientific integrity.

These three issues are mandatory requirements for acceptance. The remaining points raised by the reviewer have been adequately addressed or can be considered optional improvements.

Upon resubmission, provide a revised manuscript with tracked changes clearly showing all modifications, a clean version, and a detailed response letter citing specific line numbers for each change.

Please understand that this is my final decision as Handling Editor. If these three critical requirements are not satisfactorily met in your next submission, unfortunately, the manuscript will be rejected.

Hermosillo, Sonora, January 07, 2025
Dr. Ernesto Caffarena
Handling Editor
Memórias do Instituto Oswaldo Cruz

Dear Dr. Caffarena,
Enclosed please find the revised version of our manuscript entitled "Immunoinformatic analysis for identifying immunogenic antigens from the complete proteome of Giardia lamblia," previously submitted to Memórias do Instituto Oswaldo Cruz under ID MIOC-2025-0216.R1

The manuscript has been carefully revised in accordance with your comments, and all of your suggestions have been fully addressed. All changes made in the text have been highlighted in bold, with the corresponding page and line numbers indicated.

We remain at your disposal for any further clarification or additional information you may require.

Sincerely,
Dr. Carlos Velazquez
Department of Chemistry-Biology
Universidad de Sonora

Responses to reviewers
Reviewer required attention about the following aspects:

Comment 1: "The reviewer made a specific recommendation to deposit your data in an open access repository and create a public code repository. You merely moved some tables to Supplementary Material, which does not constitute open data sharing. This represents a fundamental failure to comply with Open Science principles that are non-negotiable for publication in this journal. You must create a public repository (GitHub, GitLab, or equivalent) containing all analysis code, peptide prediction datasets, and processing scripts. Additionally, deposit your complete datasets in a recognized repository such as Zenodo, Figshare, or Dryad with a DOI. Include all repository URLs and DOI numbers in your Data Availability Statement."

Answer 1: We thank the reviewer for the helpful feedback and agree with the proposed suggestion. A public repository hosted on Zenodo and GitHub has been created in which the analysis code, scripts, and all datasets have been deposited. The repository URLs and DOI numbers have been included in the Data Availability Statement (page 18, lines 406-410), as requested.

GitHub URLs: https://github.com/davidortega47-rgb/Datasets_Immunoinformatic_Prediction_G.lamblia.git
https://github.com/davidortega47-rgb/LIBCE-codes.git
Zenodo DOI numbers:
https://doi.org/10.5281/zenodo.18166891
https://doi.org/10.5281/zenodo.18166880

Comment 2: "The reviewer raised concerns about your overly restrictive cut-off thresholds potentially missing relevant epitopes. Your justification based on practical convenience is scientifically insufficient. You must perform a sensitivity analysis by re-running your predictions using at least two additional thresholds (e.g., IC50 <100 nM and <500 nM, or percentile rank <0.1 and <1.0). Present a summary table in the supplementary material showing: (a) the total number of candidate polypeptides identified at each threshold, (b) the number of these showing promiscuous binding, and (c) overlap between threshold groups. If experimentally validated Giardia lamblia T-cell epitopes are available in the published literature, demonstrate whether your thresholds capture them; if no such epitopes exist, explicitly state this limitation. This analysis will allow assessment of the robustness of your candidate selection strategy."

Answer 2: We are grateful for the reviewer's thoughtful remarks and accept the suggestions made. Predictions were re-run using two additional thresholds (percentile rank < 0.1, IC50 < 50 nM; and percentile rank < 1, IC50 < 50 nM) in order to generate a summary table and compare predictions across the different thresholds. This table is mentioned in the Discussion (page 16, lines 361-368) and will be provided as supplementary material. Limitations regarding the availability of experimentally validated Giardia lamblia T-cell epitopes are also addressed in the Discussion (page 16, lines 371-374).

Comment 3: "The reviewer explicitly stated that it must be made clear throughout the text that GiardiaDB protein sequences are unvalidated computational predictions. Five lines buried in the Discussion is insufficient for a limitation that affects the validity of your entire dataset. You must add explicit statements in the Introduction when

first presenting GiardiaDB as your data source, in the Methods section when describing protein sequence retrieval, and in your Conclusions when presenting candidate antigens. This level of transparency about data quality is fundamental to scientific integrity."

Answer 3: We appreciate the reviewer's valuable feedback and concur with the observation. Explicit statements regarding the limitation of GiardiaDB related to non-validated sequences have been added to the Introduction (page 6, lines 126-132), Methodology (page 6, lines 139-141), and Conclusions (page 15, lines 352-355), as previously suggested.

## THIRD REVIEW ROUND

REVIEWERS' COMMENTS

**REVIEWER #1**

No other comments.

