## [Reviewer Report · FIRST REVIEW ROUND - REVIEWERS COMMENTS]

## REVIEWER #1

The abstract clearly states the rationale (lack of human vaccine), the aim (systematic immunoinformatic screen), methodology (NetMHCII 2.3, murine MHC-II alleles, Python pipeline), main results (414 immunogenic WB, 350 GS, conserved/promiscuous polypeptides, secretome hits), and significance (first comprehensive proteome-wide map). It is concise, well-structured, and understandable for a broad readership. However, it is written in a very technical language: IC50 values and percentile thresholds may be too detailed for the abstract. The clinical relevance is mentioned only in the last line; could be emphasized earlier. “Data science” is mentioned vaguely. It remains unclear what beyond the NetMHCII predictions was novel. Taken together, the structure is fine but the abstract would benefit from reducing technical jargon, and highlighting the potential translational impact more clearly.

The study is original and the claim to be the first complete proteome-wide immunoinformatic screen for *Giardia* is valid and novel. It provides a foundational catalog of candidate immunogens, which is highly relevant given the absence of a licensed vaccine and the burden of giardiasis globally. However, the translational step from murine MHC-II predictions to human HLA applicability is somewhat underdeveloped, although parallels with HLA-DR are discussed. Most importantly, the fact that experimental validation is absent limits the impact of this work. Hence, conclusions about vaccine candidacy should be toned down.

The authors rely on a single data source (GiardiaDB proteomes, secretome datasets) which is appropriate but requires explicit consideration of limitations. While the genome assemblies in GiardiaDb are robust, automated annotation and gene model prediction is not validated. It has to be made clear in the text that protein sequences are unvalidated predictions and only as good as the gene models are.

The use of NetMHCII 2.3 and stringent cutoffs (≤0.01 percentile, IC50 ≤ 50 nM) provides rigor, though may be overly restrictive, missing known epitopes. Homology and secretome integration are well described. A limitation is that only murine alleles were tested. The lack of direct HLA predictions weakens translational relevance.

The results are comprehensive, with clear reporting of numbers of immunogenic proteins, conserved homologs, secretome overlaps, and promiscuous peptides. There are too many tables and figures. I recommend organizing and moving this data to deposit in an open access repository and make the links available in a newly created Gitlab repository for the project.

Albeit strong in contextualizing findings with immunobiology of *Giardia*, MHC binding properties, and parallels to human immunity, the discussion is exceedingly long and needs to be reorganized to serve its direct purpose and significantly shortened. The authors should appropriately acknowledge the limitations of prediction tools, the need for in vivo validation, and the missed known epitopes.

## AUTHORS’ RESPONSE TO THE REVIEWERS

Hermosillo, Sonora, November 10th, 2025

Dr. Ernesto Caffarena

Handling Editor

Memórias do Instituto Oswaldo Cruz

Dear Dr. Caffarena,

Enclosed please find the revised version of our manuscript entitled “Immunoinformatic analysis for identifying immunogenic antigens from the complete proteome of *Giardia lamblia*,” previously submitted to Memórias do Instituto Oswaldo Cruz under ID MIOC-2025-0216.

The manuscript has been carefully revised according to the reviewers’ comments, and this document provides a detailed, point-by-point response to each of their observations.

We remain at your disposal for any further clarification or additional information you may require.

Sincerely,

Dr. Carlos Velazquez

Department of Chemistry-Biology

Universidad de Sonora

Responses to reviewers

Reviewer 1

Reviewer 1 required attention about the following aspects:

**Comment 1:** “It is concise, well-structured, and understandable for a broad readership. However, it is written in a very technical language: IC50 values and percentile thresholds may be too detailed for the abstract” and “The clinical relevance is mentioned only in the last line; could be emphasized earlier. “Data science” is mentioned vaguely. It remains unclear what beyond the NetMHCII predictions was novel. Taken together, the structure is fine but the abstract would benefit from reducing technical jargon and highlighting the potential translational impact more clearly”.

**Answer 1:** We thank the reviewer for these thoughtful comments and agree with the suggestions. The clinical relevance of this work is now mentioned earlier in the abstract (page 2, lines 29-32), and the potential translational applications are described more clearly (page 2, lines 44-46). Although the terms IC50 and percentile rank are standard cut-off criteria for selecting strong-binding peptides, they are already described in the main text and have therefore been removed from the abstract. The revised abstract now refers more generally to “strong binder peptides” or “sequences with high affinity for MHC class II molecules”. In addition, we now clarify the distinction between immunoinformatics and data science in the Materials and Methods section: immunoinformatics refers to the use of epitope prediction and promiscuity tools, whereas data science refers to the use of Python 3.9 to develop a pipeline for processing and analyzing large datasets obtained from the IEDB and GiardiaDB databases. This approach allowed us to integrate and automate analyses beyond standard NetMHCII predictions, highlighting the novel computational contribution of our study.

**Comment 2:** “The study is original and the claim to be the first complete proteome-wide immunoinformatic screen for *Giardia* is valid and novel. It provides a foundational catalog of candidate immunogens, which is highly relevant given the absence of a licensed vaccine and the burden of giardiasis globally. However, the translational step from murine MHC-II predictions to human HLA applicability is somewhat underdeveloped, although parallels with HLA-DR are discussed. Most importantly, the fact that experimental validation is absent limits the impact of this work. Hence, conclusions about vaccine candidacy should be toned down”

**Answer 2:** We appreciate the reviewer’s thoughtful feedback and concur with the proposed suggestions. The manuscript has been revised to address the limitations of experimental models concerning the translation of vaccine candidates to humans (page 16, lines 371-375). The murine model is widely used in *Giardia* immunology research and provides a valuable starting point for identifying novel candidates that may confer protection against infection. Additionally, predictions based on murine MHC-II alleles can guide future studies exploring human HLA applicability. Regarding experimental validation, the ideal next step would be to select a subset of polypeptides and perform experimental analyses to confirm their immunogenicity. We have accordingly tempered the language in the conclusions to reflect these limitations and avoid overstating vaccine candidacy.

**Comment 3:** “The authors rely on a single data source (GiardiaDB proteomes, secretome datasets) which is appropriate but requires explicit consideration of limitations. While the genome assemblies in GiardiaDb are robust, automated annotation and gene model prediction is not validated. It has to be made clear in the text that protein sequences are unvalidated predictions and only as good as the gene models are¨

**Answer 3:** We thank the reviewer for this important comment and agree with the observation. GiardiaDB (https://giardiadb.org/giardiadb/app) is a repository containing genome and proteome data from several *Giardia* genotypes. Some data have been validated using microarrays and mass spectrometry, and deprecated genes have been discarded (Aurrecoechea C., 2008). However, we acknowledge that, like other biological databases (e.g., GenBank, IEDB, UniProt), GiardiaDB contains protein sequences that are computational predictions and may include redundancies or unvalidated entries. These limitations have now been explicitly acknowledged in the manuscript (pages 16-17, lines 376-380). Despite these limitations, the dataset provides a comprehensive starting point for identifying potential immunogens and guiding future experimental validation.

**Comment 4:** “The use of NetMHCII 2.3 and stringent cutoffs (≤0.01 percentile, IC50 ≤ 50 nM) provides rigor, though may be overly restrictive, missing known epitopes”

**Answer 4:** We appreciate the reviewer’s valuable feedback and concur with the observation. Commonly used affinity-based thresholds for T cell epitopes are IC50 < 500 nM and percentile rank < 10% (Reardon B., 2021; Paul S., 2013), corresponding to approximately 80% sensitivity in predicting true ligands. By applying more stringent cut-off points (IC50 < 50 nM and rank < 0.01), we aimed to select the most immunogenic antigens from the ~4,500 proteins comprising the *Giardia* proteome. We initially evaluated less stringent cut-off points (rank < 0.1, rank < 1, rank < 5), but these yielded over 1,000 polypeptides per genotype, which was impractical for downstream analyses. We acknowledge that these stringent thresholds may exclude some epitopes, constituting a limitation of the study, which is discussed in the manuscript (page 15, lines 347-357). Nevertheless, this approach ensures that the selected candidates are highly likely to be immunogenic, providing a focused set of targets for future experimental validation.

**Comment 5:** “There are too many tables and figures. I recommend organizing and moving this data to deposit in an open access repository and make the links available in a newly created Gitlab repository for the project”

**Answer 5:** We thank the reviewer for the helpful feedback and agree with the proposed suggestion. Some tables have been moved to the Supplementary Material section of the revised manuscript.

**Comment 6:** “The discussion is exceedingly long and needs to be reorganized to serve its direct purpose and significantly shortened. The authors should appropriately acknowledge the limitations of prediction tools, the need for in vivo validation, and the missed known epitopes”

**Answer 6:** We are grateful for the reviewer’s thoughtful remarks and accept the suggestions made. General topics, such as *Giardia* immunobiology and MHC properties, have been abbreviated to streamline the Discussion. The limitations of using prediction tools, including the potential to miss known epitopes and the necessity for in vivo validation, have been explicitly acknowledged in the revised Discussion section (pages 15-16, lines 344-375).

---

## [Reviewer Report · REVIEWERS COMMENTS]

## REVIEWER #1

Manuscript ID MIOC-2025-0216.R1 entitled “Immunoinformatic analysis for identifying immunogenic antigens from the complete proteome of *Giardia lamblia*” which you submitted to the Memórias do Instituto Oswaldo Cruz, has been fully evaluated again, and some concerns were raised about the manuscript as it currently stands. These issues must be addressed before we would be willing to consider a revised version of your study.

Since the original reviewer is unavailable, I have personally assumed responsibility for this assessment. After thorough examination, I must inform you that the manuscript still requires substantial revisions before it can be reconsidered for publication.

While I acknowledge your efforts to address several of the reviewer’s concerns, three critical issues require substantial additional work before acceptance.

First and most critically, you have completely failed to address the reviewer’s Comment 5 regarding open data accessibility. The reviewer made a specific recommendation to deposit your data in an open access repository and create a public code repository. You merely moved some tables to Supplementary Material, which does not constitute open data sharing. This represents a fundamental failure to comply with Open Science principles that are non-negotiable for publication in this journal. You must create a public repository (GitHub, GitLab, or equivalent) containing all analysis code, peptide prediction datasets, and processing scripts. Additionally, deposit your complete datasets in a recognized repository such as Zenodo, Figshare, or Dryad with a DOI. Include all repository URLs and DOI numbers in your Data Availability Statement. Without this, the manuscript cannot be accepted.

Second, the reviewer raised concerns about your overly restrictive cut-off thresholds potentially missing relevant epitopes. Your justification based on practical convenience is scientifically insufficient. You must perform a sensitivity analysis by re-running your predictions using at least two additional thresholds (e.g., IC50 <100 nM and <500 nM, or percentile rank <0.1 and <1.0). Present a summary table in the supplementary material showing: (a) the total number of candidate polypeptides identified at each threshold, (b) the number of these showing promiscuous binding, and (c) overlap between threshold groups. If experimentally validated *Giardia lamblia* T-cell epitopes are available in the published literature, demonstrate whether your thresholds capture them; if no such epitopes exist, explicitly state this limitation. This analysis will allow assessment of the robustness of your candidate selection strategy.

Third, regarding Comment 3, the reviewer explicitly stated that it must be made clear throughout the text that GiardiaDB protein sequences are unvalidated computational predictions. Five lines buried in the Discussion is insufficient for a limitation that affects the validity of your entire dataset. You must add explicit statements in the Introduction when first presenting GiardiaDB as your data source, in the Methods section when describing protein sequence retrieval, and in your Conclusions when presenting candidate antigens. This level of transparency about data quality is fundamental to scientific integrity.

These three issues are mandatory requirements for acceptance. The remaining points raised by the reviewer have been adequately addressed or can be considered optional improvements.

Upon resubmission, provide a revised manuscript with tracked changes clearly showing all modifications, a clean version, and a detailed response letter citing specific line numbers for each change. Please understand that this is my final decision as Handling Editor. If these three critical requirements are not satisfactorily met in your next submission, unfortunately, the manuscript will be rejected.

## AUTHORS’ RESPONSE TO THE REVIEWERS

Hermosillo, Sonora, January 07, 2026

Dr. Ernesto Caffarena

Handling Editor

Memórias do Instituto Oswaldo Cruz

Dear Dr. Caffarena,

Enclosed please find the revised version of our manuscript entitled “Immunoinformatic analysis for identifying immunogenic antigens from the complete proteome of *Giardia lamblia*,” previously submitted to Memórias do Instituto Oswaldo Cruz under ID MIOC-2025-0216.R1

The manuscript has been carefully revised in accordance with your comments, and all of your suggestions have been fully addressed. All changes made in the text have been highlighted in bold, with the corresponding page and line numbers indicated.

We remain at your disposal for any further clarification or additional information you may require.

Sincerely,

Dr. Carlos Velazquez

Department of Chemistry-Biology

Universidad de Sonora

Responses to reviewers

Reviewer required attention about the following aspects:

**Comment 1:** “The reviewer made a specific recommendation to deposit your data in an open access repository and create a public code repository. You merely moved some tables to Supplementary Material, which does not constitute open data sharing. This represents a fundamental failure to comply with Open Science principles that are non-negotiable for publication in this journal. You must create a public repository (GitHub, GitLab, or equivalent) containing all analysis code, peptide prediction datasets, and processing scripts. Additionally, deposit your complete datasets in a recognized repository such as Zenodo, Figshare, or Dryad with a DOI. Include all repository URLs and DOI numbers in your Data Availability Statement.”

**Answer 1:** We thank the reviewer for the helpful feedback and agree with the proposed suggestion. A public repository hosted on Zenodo and GitHub has been created in which the analysis code, scripts, and all datasets have been deposited. The repository URLs and DOI numbers have been included in the Data Availability Statement (page 18, lines 406-410), as requested.

GitHub URLs: https://github.com/davidortega47-rgb/Datasets_Immunoinformatic_Prediction_G.lamblia.git

https://github.com/davidortega47-rgb/LIBCE-codes.git

Zenodo DOI numbers:

https://doi.org/10.5281/zenodo.18166891

https://doi.org/10.5281/zenodo.18166880

**Comment 2:** “The reviewer raised concerns about your overly restrictive cut-off thresholds potentially missing relevant epitopes. Your justification based on practical convenience is scientifically insufficient. You must perform a sensitivity analysis by re-running your predictions using at least two additional thresholds (e.g., IC50 <100 nM and <500 nM, or percentile rank <0.1 and <1.0). Present a summary table in the supplementary material showing: (a) the total number of candidate polypeptides identified at each threshold, (b) the number of these showing promiscuous binding, and (c) overlap between threshold groups. If experimentally validated *Giardia lamblia* T-cell epitopes are available in the published literature, demonstrate whether your thresholds capture them; if no such epitopes exist, explicitly state this limitation. This analysis will allow assessment of the robustness of your candidate selection strategy.”

**Answer 2:** We are grateful for the reviewer’s thoughtful remarks and accept the suggestions made. Predictions were re-run using two additional thresholds (percentile rank < 0.1, IC50 < 50 nM; and percentile rank < 1, IC50 < 50 nM) in order to generate a summary table and compare predictions across the different thresholds. This table is mentioned in the Discussion (page 16, lines 361-368) and will be provided as supplementary material. Limitations regarding the availability of experimentally validated *Giardia lamblia* T-cell epitopes are also addressed in the Discussion (page 16, lines 371-374).

**Comment 3:** “The reviewer explicitly stated that it must be made clear throughout the text that GiardiaDB protein sequences are unvalidated computational predictions. Five lines buried in the Discussion is insufficient for a limitation that affects the validity of your entire dataset. You must add explicit statements in the Introduction when first presenting GiardiaDB as your data source, in the Methods section when describing protein sequence retrieval, and in your Conclusions when presenting candidate antigens. This level of transparency about data quality is fundamental to scientific integrity.¨

**Answer 3:** We appreciate the reviewer’s valuable feedback and concur with the observation. Explicit statements regarding the limitation of GiardiaDB related to non-validated sequences have been added to the Introduction (page 6, lines 126-132), Methodology (page 6, lines 139-141), and Conclusions (page 15, lines 352-355), as previously suggested.